# Pediatric Experience and Outcomes from the First Single-Vault Compact Proton Therapy Center

**DOI:** 10.3390/cancers15164072

**Published:** 2023-08-12

**Authors:** Stephanie M. Perkins, Sabrina Prime, Michael Watts, Jiayi Huang, Tianyu Zhao

**Affiliations:** S. Lee Kling Proton Therapy Center, Washington University School of Medicine/Siteman Cancer Center, Saint Louis, MO 63110, USA

**Keywords:** proton therapy, pediatric, compact, single-room, socioeconomic

## Abstract

**Simple Summary:**

Protons have unique physical attributes that allow for a reduction in normal tissue dose when treating patients with cancer. For these reasons, proton therapy is of benefit for many children undergoing radiation therapy due to their sensitivity to ionizing radiation. Prior to 2013, only multi-room large proton therapy centers were available with very high cost and space requirements. Here, we sought to evaluate the pediatric experience and survival outcomes at the world’s first single-vault compact center that opened at our institution in 2013. We assessed the demographics of our patients, diagnosis types, treatment approach, need for anesthesia, distance travelled, and increase in our pediatric service by outside referrals for proton therapy.

**Abstract:**

The first single-vault compact proton therapy center opened in 2013, utilizing a gantry-mounted synchrocylotron. The center was placed within a large academic radiation oncology department with a high priority for pediatric cancer care. Here we performed a retrospective study of pediatric (≤21 years) patients treated with proton therapy at our institution between 2013–2022. Patient, tumor, and treatment characteristics were obtained including race, socioeconomic status, insurance type, distance travelled, need for anesthesia, and outside referrals for proton therapy. In total, 250 pediatric patients were treated with proton therapy comprising 18% of our proton patient volume. Median follow-up was 3.1 years, 38.4% were female and 83% were white. The majority of cases were CNS (69.6%) and a large number of patients (80/250, 32%) required craniospinal irradiation. Anesthesia was required for 39.6% of patients. Average distance travelled for treatment was 111 miles and 23% of patients were referred from outside institutions for proton therapy. Insurance type was private/commercial for 61.2% followed by Medicaid for 32%. We found that 23% of patients lived in census tracts with >25% of people living below the national poverty line. Overall survival at 3 years was excellent at 83.7% with better outcomes for CNS patients compared to non-CNS patients. There were no cases of secondary malignancy at this early time point. As the world’s first compact proton therapy center, we found that proton therapy increased our pediatric volume and provided proton therapy to a diverse group of children in our region. These data highlight some of the expected patient and tumor characteristics and necessary resources for providing pediatric proton beam therapy.

## 1. Introduction

Proton radiation therapy offers several unique physical and biological properties which enable the delivery of high-dose radiation to sensitive areas, while minimizing normal tissue toxicities by way of highly conformed treatment plans. This degree of conformality is owed largely to the inherent Bragg peak of the positively charged particle, which deposits the majority of its dose within the clinical target volume, as opposed to the front-heavy dose deposition of photons which decreases with depth. These physical attributes allow for greater normal tissue sparing with proton therapy and thus proton therapy is widely used in the treatment of pediatric malignancies due to the importance of limiting radiation to healthy tissues in children [1,2,3,4,5]. Prior to 2013, all proton therapy centers were multi-vault centers requiring a large amount of space for construction and costs exceeding USD 100 million for most centers. Due to these factors, in 2010, only about 10 treatment centers were available in the United States—despite this technology being in use since the 1970s. 

In 2013 our institution installed the first single-room compact proton therapy center at significantly lower cost than a traditional multi-room center [6]. The single-room solution allowed for proton therapy availability at our main campus—a congested urban site—in which a large multi-room center was impossible. The proton center would function as an extension of our main department and was located adjacent to our pediatric hospital, ensuring convenient accessibility for the treatment of our pediatric inpatients. Since the opening of our center in 2013, the landscape for proton therapy centers has changed significantly. There are now 42 operational centers in the United States [7]. Additionally, a growing number of proton therapy vendors now provide the option of a single room and more compact solutions, as most centers planned at this time will accommodate 1–2 rooms for treatment. This has considerably lowered the cost of proton therapy centers and increased availability for patients throughout the United States.

To date, we have treated nearly 1500 patients with proton therapy. Here, we sought to specifically characterize and evaluate the pediatric volume of cases in a large academic medical institution with a compact proton therapy center. We evaluated the diagnoses treated, outside referrals, and distance travelled for our pediatric patients. We assessed the need for anesthesia with emphasis that our program focuses highly on the reduction of anesthesia through child life therapists and audio-visual aids. Lastly, we evaluated our long-term disease outcomes for patients treated with our proton center’s single-room synchrocylotron technology. 

## 2. Materials and Methods

This is a single-institution, retrospective study of all pediatric patients, ≤21 years old, who underwent proton therapy at the S. Lee Kling Proton Therapy Center in Saint Louis, MO. The study was approved by the Washington University Institutional Review Board. Patients were treated between the years 2014 and 2022. Data related to patient demographics, diagnosis, and disease-specific and overall outcomes were obtained. Socioeconomic status as defined by the percentage of people living below the poverty level in the patient’s census tract was collected for each patient. Additionally, the distance from the proton center to the patient’s home was calculated. Insurance type was collected for all patients and categorized as in-state Medicaid, out-of-state Medicaid, commercial/private, Tricare/military, or other. 

Between 2013 and 2020, patients were treated with passive scatter proton therapy utilizing the Mevion S250 proton therapy system (Mevion Medical Systems, Inc., Littleton, MA, USA). Beginning in June of 2020 all patients were treated with pencil-beam scanning technology utilizing the Mevion HYPERSCAN system. Daily imaging for set-up consisted of kV imaging with the option for mobile CT for patients treated with passive scatter. In-room CT-on-rails, in addition to kV imaging, was available for patients treated with pencil-beam scanning. CT simulation was performed in the main radiation oncology department. When necessary, anesthesia was provided by a pediatric anesthesia team from St. Louis Children’s Hospital that is designated to radiation oncology during the morning hours of each weekday. Propofol sedation was utilized without the use of laryngeal mask airway or intubation except in very rare scenarios. Patients that needed daily sedation were required to have PICC, Broviac, or Port-a-cath access during their course of radiation. Patients that required inpatient care were transferred to the radiation oncology department via wheelchair or hospital bed from St. Louis Children’s Hospital to the proton center.

In order to ease anxiety and decrease the need for anesthesia, the proton center has a full-time child life specialist that meets with the patient and their families at the time of consult. During that visit, an instructional video is provided to children that may be able to complete treatment awake. This video describes the process of simulation (i.e., making immobilization mask) and treatment. Beginning in 2019, the proton center utilized the audiovisual-assisted therapeutic ambience in RT (AVATAR) system that allowed children to watch videos during their treatments [8,9]. 

The primary objective of this study was to examine and characterize the pediatric proton experience at our institution through several measures related to our patients’ demographics, the types of tumors treated, the required use of pediatric anesthesia/sedation, and our treatment outcomes to date. Overall survival was measured from the first date of radiation therapy to the date of their last available follow-up or death. All patients were followed as per usual care or per study protocol if enrolled in a clinical trial. Data were carefully collected for the incidence of secondary malignancies. 

All statistical analyses were performed using SPSS Statistical Software version 27 (SPSS Inc., Chicago, IL, USA). Descriptive statistics were performed and reported baseline patient and disease characteristics. The Kaplan–Meier method was used to obtain overall survival (OS) and disease-specific survival (DSS) curves and review event–time distributions. Given the descriptive nature of this report and the large heterogeneity of tumor types reported, no statistical comparison tests were used. 

## 3. Results

Between 2013 and 2022, 250 pediatric patients received proton beam radiation therapy at our center. The proton center treated 1385 patients during this time and thus pediatric cases comprised 18% of our patient volume. Patient demographic information is presented in Table 1. The median age at time of treatment was 10.9 years old (range 8 months–21 years), and 38.4% were female. Infants (as defined as age < 3 years old), comprised 8.8% of patients. On average, patients travelled 111 miles (range 2–438) for proton therapy (excluding two international patients from Europe), with 21% of patients travelling > 200 miles for proton therapy. In total, 58 (23.2%) patients were referred to our institution from outside hospitals solely for proton therapy. The average distance travelled for these outside referrals was 188 miles (range 9–438). Regarding socioeconomic status, 23% of patients lived in census tracts with >25% of people living below the national poverty line. Insurance type as detailed in Table 1 shows that the majority of children (61.2%) had private/commercial insurance followed by in-state Medicaid, out-of-state Medicaid, and Tricare. For the entire cohort, 12% of patients were black and 83% were white. Looking at our local children, as defined as distance travelled of 0–40 miles, we found that 28% of our patients were black compared to 8.6% for our outside referral patients.

Details regarding tumor type and treatment characteristics are provided in Table 2. Central nervous system (CNS) tumors were the most common and comprised 68.9% of all cases. Within the CNS cases, medulloblastoma was most common, followed by ependymoma. Intracranial reirradiation accounted for 8/174 (4.6%) of CNS cases due to disease recurrence. Non-CNS tumors accounted for 31.1% of all cases, with the most common being rhabdomyosarcoma followed by lymphoma. 

For the entire cohort, anesthesia was required for 99 (39.6%) patients with a median age of 5.1 years (range 8 months–14 years). The youngest age where anesthesia was not required was 4.2 years. Anesthesia was utilized commonly for craniospinal irradiation (CSI) (50/80 [62.5%]) patients. The youngest awake craniospinal case was 10.6 years old. CSI was a common treatment across our cohort with 80/250 (32%) patients requiring it. Passive scatter proton therapy was used for 179 (71.6%) patients and pencil beam scanning technology was used for 71 (28.4%). The type of proton therapy utilized was determined by treatment date, as our center moved from passive scatter to pencil beam scanning in 2020.

Median follow-up for the cohort was 3.1 years (range 0.17–8.4). At the time of this analysis, 200 patients were living, representing 83.7% of the cohort. Thirty-three patients had succumbed to their disease (13.8%) and an additional 6 patients (2.5%) had expired due to other causes. Of the 200 living patients, 13 were living with disease (6.5%). Overall survival (OS) at 3 and 5 years for the entire cohort was 83.7% and 81.3%, respectively (Figure 1A). Disease-free survival at 3 and 5 years was 81.1% and 78.1%, respectively (Figure 1A). CNS patients fared better than non-CNS tumor patients with a 3-year OS of 86.3% versus 77.7%, respectively (Figure 1B). 

Individual tumor types within CNS and non-CNS cites demonstrated considerable heterogeneity of outcomes. The estimated 3-year OS for CNS patients by tumor type was as follows: atypical teratoid/rhabdoid tumor (ATRT) 100%, low-grade glioma (LGG) 100%, germ-cell tumor (GCT) 93.8%, medulloblastoma 92.7%, craniopharyngioma 92.3%, ependymoma 91.8%, and other CNS tumors 67.3% (Figure 2A). The estimated 3-year OS for non-CNS patients by tumor type was: lymphoma 100%, neuroblastoma 83.3%, rhabdomyosarcoma 81.8%, other non-sarcoma tumors 66.7%, Ewing sarcoma 52.5%, and other sarcoma 51.4% (Figure 2B).

No radiation-induced secondary malignancies were observed at the time of analysis. Three patients experienced a benign lesion in the field of craniospinal radiation; these included a benign osteoma, fibrous dysplasia, and an aneurysmal bone cyst. Each was treated with either resection or biopsy alone with no adverse effects. 

## 4. Discussion

Here, we describe the pediatric experience at a large academic medical center implementing the world’s first single-vault proton therapy center. Since opening in 2013, pediatric patients have routinely comprised 17–20% of our patient volume. Given the large patient volume within our department including our satellite locations, a triage process is in place to determine the utilization of proton therapy; however, pediatric patients are the top priority for treatment and receive automatic admittance. These data provide insight into the expected patient volume, diagnosis types, age distribution, distance travelled, insurance type, and growth of service from outside referrals for proton therapy. A summary of these findings is presented in Figure 3. 

In addition to the cost advantage, the compact proton therapy design of our unit allowed for the placement of the proton center in an existing basement at our medical institution. At the time of its construction, a second vault was incorporated into the center to accommodate future expansion. Our proton center now anticipates operation as a two-vault center in the fall of 2023. From a practical perspective, the ability to locate the center adjacent to our existing department was important, as this allowed us to decrease independent physician staffing needs. Moreover, the center is connected with our adult and children’s hospitals on the Washington University Medical campus. For this study, we were unable to gather information regarding inpatient admissions during the course of proton therapy. However, we can attest that delivering proton therapy to children often overlaps with unplanned inpatient admissions—most commonly for neutropenic fever, admission for concurrent chemotherapy, need for inpatient hydration and/or pain control, or for children that were admitted to our inpatient neurological rehabilitation center for inpatient therapy services. As such, we would strongly advocate that upcoming proton therapy centers give significant consideration to its patients’ needs for potential planned or unplanned hospital admissions—especially if planning to treat pediatric patients. 

Young pediatric patients—those that arguably benefit the most from proton therapy—are unique in their radiation needs. For example, another resource consideration would include the use of anesthesia. Our center has been proactive in reducing anesthesia needs through the use of child life therapy, an education/play room for introduction to radiation therapy and simulation processes, the use of educational videos of prior patients receiving radiation, and with the use of audio-visual aids during the course of treatment. With this approach, we have reduced our anesthesia needs; however, we report that, despite these measures, 40% of our children still required anesthesia during their proton therapy course. A portion of this is attributable to the high volume of CNS cases and use of craniospinal radiation which requires more cooperation due to the length of the spinal field and increased time for treatment. In some cases, children required sedation for the craniospinal treatment, and then were able to complete their boost fractions without sedation. Overall, our findings regarding anesthesia/sedation were in line with those of other studies [10]. Regarding craniospinal treatments, we found 1 in 3 children treated at our center required craniospinal irradiation, a much higher percentage than we anticipated. Therefore, proton centers planning to treat children will need a reliable workflow in place for proton craniospinal treatments which presents unique challenges for field matching and set-up. 

Our data indicate that proton therapy allowed for the growth of our pediatric radiotherapy service with an increase in referral of pediatric patients from outside institutions. We found that 23.2% of patients were referred from outside facilities solely for proton therapy. Most children returned to their home institution upon completion of proton radiation and we gathered follow-up data from their primary local team. This large percentage of outside referrals (23.2%) is reflected in our distance travelled, but we also found that several children within our local catchment area were referred for proton therapy from neighboring institutions. During the time period of this study, our center was the only proton therapy center within 4 h of St. Louis. For prospective proton centers with similar regional size and lack of proton therapy within a 3–4 h radius, a 15–25% estimated increase in pediatric volume would be reasonable. 

In line with other proton therapy centers, we found that infants (children < 3 years of age) were commonly treated at our center and comprised 9% of our patients. These were most commonly patients with ependymoma, atypical teratoid/rhabdoid tumors, and neuroblastoma. The University of Florida proton group has reported that nearly 22% of their patients were under 3 years of age, and younger age is associated with increased utilization of proton therapy, indicating infant patients are more often referred to proton therapy centers [11,12]. Overall, CNS tumors represented the majority of tumors, with medulloblastoma being the most common tumor type. These findings were similar to those of others [13]. 

It is known that, while the use of proton therapy for children is increasing, there are concerning findings that socioeconomic factors play a role in utilization [12,14,15]. This may be due to the need for travel for many families, and often families are unable to travel due to job security, cost, or care required for other children at home. This highlights the potential for compact proton therapy centers to impact the availability for proton therapy as these centers grow throughout the United States. Our center is located in the city of St. Louis, Missouri, and we found that the average distance from home to our center, excluding two international patients, was 111 miles. Data from the National Cancer Center Database (NCDB) have shown that nearly 25% of pediatric patients will travel >100 miles for proton therapy, with 13% of patients travelling > 200 miles [14]. Proton centers must plan ahead to consider their resources for housing. Our catchment area includes the urban area of St. Louis but extends to the surrounding counties of St. Louis and into the rural areas of central Missouri and Southern Illinois. For the children from our local catchment area (<41 miles from the proton center) we found that nearly 30% of our patients were black, well over the national average for proton utilization. We found that 12.5% of patients lived in census tracts with >25% of people living below the poverty line. These patients represented families living in the city of St. Louis along with rural regions of Missouri and Illinois. Thirty-two percent (32%) of our patients had state-funded Medicaid including 21.6% in-state (Missouri) and 10.4% out-of-state, which was largely children from Illinois. Regarding the commercial/private insurance patients, analysis of the insurance approval process was beyond the scope of this paper. However, we can state that, for children 18 years of age and younger, we can recall no denials for proton therapy, although peer-to-peer was required in <5 cases and one case required a demand for pre-determination that involved the parents and the parent’s employer. For patients >18 years of age, the approval for proton therapy varied across private payors and sometimes requires peer-to-peer discussions, appeal of denials, and/or plan comparisons. The relative ease for the approval of proton therapy for children contrasts with the more onerous process of proton approval for adult patients [16]. 

As the first center using the Mevion Medical Systems gantry-mounted synchrocyclotron technology, our findings demonstrate excellent outcomes for our patients. Overall and disease-free survival were >85% at 3 years. CNS patients fared better than non-CNS patients largely due to poorer outcomes in high-risk neuroblastoma patients and in Ewing sarcoma patients—many of whom had stage IV disease at diagnosis. Although our numbers were small, we noted that with a mean follow-up of 2.4 years, 100% of patients with ATRT (N = 6) are alive with no evidence of disease, supporting the use of proton therapy for these young patients. A limitation of this study is that we did not retrospectively access toxicity for our patients. However, we have previously published proton data regarding brainstem toxicity with no cases of symptomatic brain stem toxicity in our patients [17]. Given the neutrons produced during passive scatter proton therapy, attention to secondary malignancy risk is of interest. In our data set, there were no secondary malignancies. However, our median follow-up was 3.1 years, which is an early time point for this assessment. A longer follow-up will be needed to truly assess this risk. However, our findings are in line with others thus far in that second tumor risks during the first 5–10 years after proton therapy treatment are rare. In a report of 1713 children treated with double scatter proton therapy, the 5- and 10-year cumulative incidence of secondary tumors was 0.8% and 3.1%, respectively, at a median follow-up of 3.3 years (range 0.1–12.8) [11]. 

## 5. Conclusions

In summary, as the world’s first compact proton therapy center we found that proton therapy increased our pediatric volume through increases in outside referrals to our center and allowed us to provide proton therapy for children in our local and regional catchment area from diverse racial and socioeconomic backgrounds. As a single-room center in a large academic radiation oncology department, pediatric cases comprised 18% of our case volume. The majority of cases were brain tumors and 40% of our patients required sedation for treatments. These data highlight some of the expected patient and tumor characteristics and indicate the need for consideration for anesthesia, housing, and travel resources for children receiving proton beam radiation therapy in the United States. These data highlight our early outcomes data and this cohort will continue to be followed for long-term analysis in the future. 

## Figures and Tables

**Figure 1 cancers-15-04072-f001:**
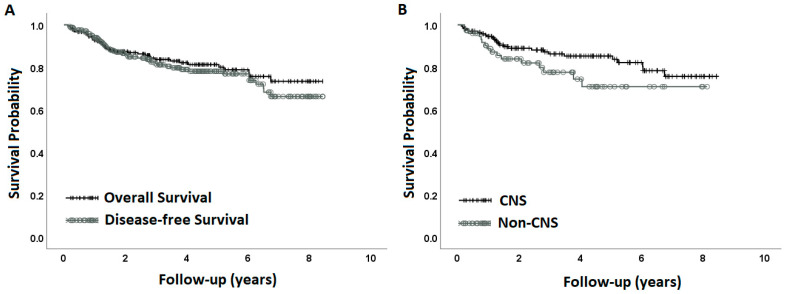
Overall survival for the entire cohort and disease-free survival for the entire cohort (**A**). Overall survival is presented for CNS patients and non-CNS patients (**B**).

**Figure 2 cancers-15-04072-f002:**
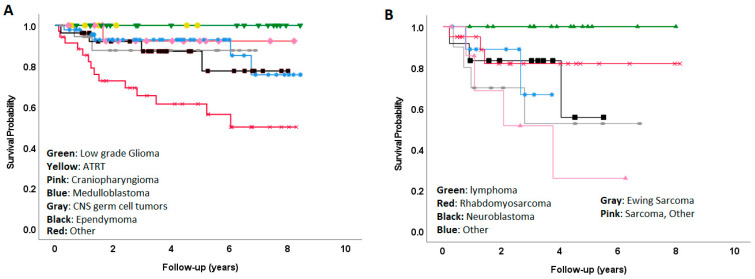
Overall survival by individual diagnosis for CNS (**A**) and non-CNS (**B**) tumors.

**Figure 3 cancers-15-04072-f003:**
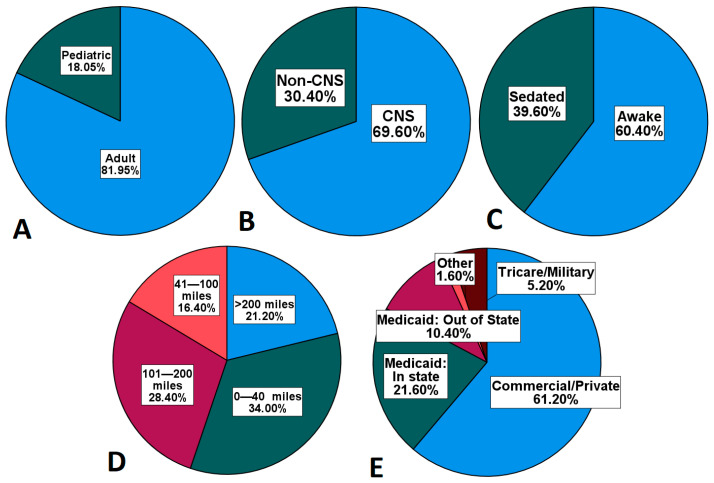
Percentage of pediatric patients (**A**) by diagnosis type (**B**), need for sedation (**C**), distance travelled for proton therapy (**D**), and insurance type (**E**).

**Table 1 cancers-15-04072-t001:** Patient characteristics.

Patient Characteristics	N (%)
**Sex**	
Male	154 (61.6)
Female	96 (38.4)
**Race**	
White	209 (83.6)
Black	30 (12.0)
Multiple Races/Other	11 (4.4)
**Age**	
0–2 years	22 (8.8)
3–7 years	67 (26.8)
8–11 years	52 (20.8)
12–15 years	53 (21.2)
16–21 years	56 (22.4)
**SES status (% living below poverty level in census tract)**	
0–10%	102 (41.1)
10–25%	115 (46.4)
>25%	31 (12.5)
**Distance traveled for treatment (miles)**	
0–40	85 (34.0)
41–100	41 (16.5)
101–200	71 (28.4)
>200	53 (21.2)
**Outside referrals for proton therapy**	58 (23.2)
**Insurance type**	
Private/Commercial	153 (61.2)
Medicaid: In-state	54 (21.6)
Medicaid: Out-of-state	26 (10.4)
Tricare/Military	13 (5.2)
Other	4 (1.6)

Abbreviations: SES = socioeconomic status as defined as the percentage of people living below the poverty line in the patient’s census tract (two international patients not included).

**Table 2 cancers-15-04072-t002:** Diagnosis and treatment characteristics.

	N (%)
**Treatment Site**	
CNS	174 (69.6)
Non-CNS	76 (30.4)
**CNS Tumor Types (N = 174)**	
Medulloblastoma	46 (26.4)
Ependymoma	27 (15.5)
CNS GCT	20 (11.5)
Low-grade glioma	18 (10.3)
Craniopharyngioma	17 (9.8)
ATRT	6 (3.4)
Other	40 (22.9)
**Non-CNS Tumor Types (N = 76)**	
Rhabdomyosarcoma	20 (26.3)
Neuroblastoma	12 (15.8)
Lymphoma	15 (19.7)
Ewing Sarcoma	11 (14.5)
Sarcoma, other	8 (10.5)
Other	10 (13.2)
**Patients requiring Craniospinal**	80 (32.0)
**Proton Modality**	
Passive Scatter	179 (71.6)
Pencil Beam Scanning	71 (28.4)
**Patients requiring anesthesia**	99 (39.6)

Abbreviations: CNS = central nervous system, GCT = germ cell tumor, ATRT = atypical teratoid/rhabdoid tumor.

## Data Availability

The data presented in this study are available in this article.

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
