# Peer review of "Pediatric Experience and Outcomes from the First Single-Vault Compact Proton Therapy Center"

_cancers, 2023, doi:10.3390/cancers15164072_

Round 1

Reviewer 1 Report

In this work, the authors present their findings and experience in treating patients with proton therapy at a single room proton therapy center. They present patient outcomes (broken up by disease site) and provide insight for others that are considering pursuing a single room proton therapy unit (e.g. to anticipate a high number of CSI treatments; explaining and referencing the use of AVATAR to reduce the use of anesthesia in treating pediatric patients).  The work presented was approved by the institution's IRB; the figures are all clear with the exception being Figure 1 where the authors may consider making changes to improve the readability of the figure.  The language is clear (minor typographical error on line 82 should be "kV" and not "Kv"). 

Overall, this paper is of interest to the community, well written and clearly presented. 

Author Response

Thank you for the supportive comments and suggested edits.  Kv was edited to kV on line 82.  Text size enlarged on Figure 1 to improve readability. 

Reviewer 2 Report

I am grateful for the opportunity to review manuscript ID cancers-2525576, entitled “Pediatric experience and outcomes from the first single-vault compact proton therapy center.”

The authors present a well written manuscript describing their early experience of pediatric proton therapy, delivered by the world’s first compact system. The cohort consists of 250 pediatric patients treated between 2013-2022. Treatments prior to 2020 used passively scattered beams, whilst subsequent treatment were delivered by proton pencil beam scanning. Patient outcomes are reported, with median follow-up of 3.1 years (range 0.17-8.4).

Results are clearly presented, and the discussion is balanced.  I have only a few comments ahead of recommending publication:

·        First and foremost this is a report of outcomes for a cohort of pediatric patients after having received proton therapy. As such, the findings are not novel, but are consistent with similar reports in the literature. However, what is unique is the discussion around compact systems providing access to a diverse population of varied socioeconomic backgrounds with comparable clinical outcomes as those reported from the larger proton therapy facilities. It is this angle that will justify publication.

·        Following on from the previous point, the median follow-up is short. Not much can be said yet about secondary malignancies, for example. As such, I recommend that the authors highlight their rationale for reporting these data more clearly (as per my previous point), and compare to outcomes reported in the literature at similar timepoints – emphasizing that they are on track, whilst providing a more accessible serve.

·        Given these comments, I would also recommend including the word “Initial…” at the beginning of the title, or somehow indicate the early nature of the outcomes data in the title. The authors should also state their intention to continue follow-up in the discussion.

·        Some minor typos to address.

Author Response

Thank you for these supportive comments.  We agree with the suggestions regarding our early data on secondary malignancy.  We have added "at this early time-point" to the abstract.  To the discussion we have added additional information about secondary malignancy risk from other groups and further clarified that our data our early.

Reviewer 3 Report

The authors described their long-term experience (9 years) for paediatric patients treated with a compact proton synchrocyclotron with gantry. While the content is presented with small analysis due to the large heterogenity in the cohort, the paper is well written and oriented to a general clinical scientific audience. 

Author Response

Thank you for these supportive comments.

Round 2

Reviewer 2 Report

The authors have adequately addressed reviewer comments with this revised submission.

Author Response

Thank you.